# Are the Cutaneous Microbiota a Guardian of the Skin’s Physical Barrier? The Intricate Relationship between Skin Microbes and Barrier Integrity

**DOI:** 10.3390/ijms242115962

**Published:** 2023-11-04

**Authors:** Kornélia Szabó, Beáta Szilvia Bolla, Lilla Erdei, Fanni Balogh, Lajos Kemény

**Affiliations:** 1HUN-REN-SZTE Dermatological Research Group, 6720 Szeged, Hungary; 2Department of Dermatology and Allergology, Albert Szent-Györgyi Medical School, University of Szeged, 6720 Szeged, Hungary; bolla.beata.szilvia@med.u-szeged.hu (B.S.B.);; 3HCEMM-USZ Skin Research Group, 6720 Szeged, Hungary

**Keywords:** skin barrier, cutaneous microbiota, *Cutibacterium acnes*, acne vulgaris, barrier disease

## Abstract

The skin is a tightly regulated, balanced interface that maintains our integrity through a complex barrier comprising physical or mechanical, chemical, microbiological, and immunological components. The skin’s microbiota affect various properties, one of which is the establishment and maintenance of the physical barrier. This is achieved by influencing multiple processes, including keratinocyte differentiation, stratum corneum formation, and regulation of intercellular contacts. In this review, we summarize the potential contribution of *Cutibacterium acnes* to these events and outline the contribution of bacterially induced barrier defects to the pathogenesis of acne vulgaris. With the combined effects of a Westernized lifestyle, microbial dysbiosis, epithelial barrier defects, and inflammation, the development of acne is very similar to that of several other multifactorial diseases of barrier organs (e.g., inflammatory bowel disease, celiac disease, asthma, atopic dermatitis, and chronic rhinosinusitis). Therefore, the management of acne requires a complex approach, which should be taken into account when designing novel treatments that address not only the inflammatory and microbial components but also the maintenance and strengthening of the cutaneous physical barrier.

## 1. Human Skin Is a Complex Tissue with Multiple Components That Contribute to the Formation of the Cutaneous Barrier

Our skin has multiple functions. It protects us from, among other things, radiation, various chemicals and toxins, physical trauma, and harmful microbes. It regulates our body temperature, hydration and moisture levels, and senses heat and cold, touch, and pain. These properties are interrelated and structured at multiple levels. This tightly regulated and balanced interface maintains our bodies’ integrity through complex physical or mechanical, chemical, microbiological, and immunological barriers (Figure 1) [1,2]. The interconnectedness of these barriers and the importance of a tightly regulated, balanced interface between us and the environment is critical to maintaining homeostatic conditions in our bodies. Their importance is reflected in conditions where the integrity of the skin is severely compromised, such as burns or other skin disorders affecting the cutaneous barrier (e.g., different types of ichthyosis, contact and atopic dermatitis, and epidermolysis bullosa). In severe forms, the increased risk of infection and marked loss of water and ions can lead to severely decreased quality of life and even life-threatening conditions [3].

The skin is an intricate structure in which different cell types form a tissue with complex functions. This particular organ is in constant contact with the external environment and is heavily colonized by its microbes, some of which may be transient or permanent inhabitants of our bodies. Specialized bacterial, fungal, and viral species form a complex ecosystem with skin cells [4,5]. Some of them are transiently present, while others permanently populate the skin surface and various skin appendages, hair follicles within the pilosebaceous units (PSUs), and sweat glands [6,7]. These microbes and skin cells live in a very close relationship and have complex interactions. Host cells produce various factors, provide a substrate for bacterial colonization, and provide nutrients and other molecules necessary for microbial growth [8,9]. They recognize the microbes and receive information about the current state of the microbial community through direct contact and by sensing microbial molecules (e.g., metabolites, enzymes, and toxins). In turn, the microbes have a complex effect on the chemical, immunological, and microbial properties of the skin barrier [10,11], but their effect on the physical or mechanical barrier functions has been less described.

## 2. Microbes and the Physical Barrier

The structure of the human skin has evolved to support its complex functions, and several factors play a role in shaping the physical barrier functions of the skin (Figure 1). One of the most important factors is the stratum corneum (SC), which is the outermost layer of the epidermis and consists of corneocytes, which are the result of terminal differentiation processes of keratinocytes. Cornification, the process of SC formation, is a special form of programmed cell death in which the cells lose their nuclei and other organelles through the disintegration of the internal membranes, and their cytoplasm is gradually filled with densely packed fibrous keratin until they finally die [3,12].

In addition, the special properties of the deeper viable layers also contribute to this strong interface. Living keratinocytes are intimately connected across the entire width of the epidermis, so much so that the intercellular space between the cells is very small. Several types of specialized cellular constructs or intercellular junctions (desmosomes, hemidesmosomes, gap, adherent, and tight junctions—TJs) play important roles in providing such close connections [13,14,15,16,17]. The correct spatial and temporal regulation of these structures is crucial for the maintenance of a structurally and functionally mature and tight epidermal barrier. This includes, among others, the precise organization of various differentiation-related molecular components of adherens junctions and desmosomes, and the stratum granulosum-dependent formation of mature TJs [18].

Based on these data, it is not surprising that the physical barrier originates from the living epidermal layers, and balanced keratinocyte differentiation events are the key to establishing and maintaining the proper structure and barrier functions of this specialized stratified squamous epithelium [19].

The available data suggest that members of the skin microbiota have a significant impact on the different elements of the skin’s physical barrier. We summarize this knowledge in this paper (Table 1).

### 2.1. Microbes Influence Keratinocyte Differentiation

Currently, understanding of the mechanism and role of microbes in the formation of mature skin tissues is limited. During intrauterine development, a thin immature epidermis and the dermis form by 22–24 weeks of gestation, and further stratification results in five anatomically distinct epidermal layers in the third trimester. Full-term neonates already have an SC, and keratin is present in their epidermis. This structure already provides an effective shield (e.g., against water loss), but functionally mature tissues continue to develop until the first few weeks of life [20,29]. During and shortly after birth, the cutaneous microbiota begin to colonize the skin [30,31]. The timing of these events raises the possibility that in addition to assimilation to a new environment, active contact with microbes may support early neonatal development and the adaptation processes of the skin and may also help to maintain anatomically and functionally mature tissues later in life. The pH of the skin surface is 6.0 at birth and becomes slightly more acidic (pH = 5.1) during the first six weeks [20,32]; this favors the growth of the cutaneous resident microbiota instead of pathogens [33,34].

Corneocyte maturation and desquamation are regulated by the balance of various enzymes, such as lipases and proteases, and their inhibitors, as well as the physicochemical properties, such as hydration and pH, of the specific skin region [35].

In germ-free mice, relatively few SC layers are visible in electron micrographs, and their structure also appears to be different. These characteristics suggest that skin microbes affect SC formation and, therefore, the permeability barrier function [23]. Genes associated with epidermal differentiation and development show altered expression levels in the presence of skin microbes compared to mice reared under germ-free conditions. Many of the genes that have been identified belong to the epidermal differentiation complex (EDC, 1q21), including filaggrin, loricrin, involucrin, and small prolin-rich proteins, S100 proteins, which function in barrier formation [21] and skin and epidermis development [23]. In addition, immunohistochemical analysis of skin samples showed increased terminal differentiation in the epidermis in the presence of microbes, but there was increased proliferation under germ-free conditions [22].

Similar results have been reported in human in vitro keratinocyte cultures. The extracts and live and heat-killed microbes of *Cutibacterium acnes* (*C. acnes*), which is a member of the human skin microbiota, were found to increase the expression of several keratinocyte differentiation markers, including filaggrin, involucrin, and transglutaminase, suggesting that the bacterium plays an important role in maintaining an intact barrier [36,37]. In the less differentiated, suprabasal layers of ex vivo, full-thickness skin biopsy samples, filaggrin levels also increase upon treatment with a *C. acnes* extract treatment [36]. These data suggest that members of the cutaneous microbiota may have some effect on the establishment and maintenance of a mature epidermis and the development of healthy cutaneous barrier functions.

Other microbes may also affect the skin upon contact, but the effects vary among species. The extracts and heat-treated samples of *Bifidobacterium longum* (*B. longum*), which is a well-known probiotic strain, had a differentiation-promoting effect in keratinocyte cultures [38]. *Staphylococcus aureus* (*S. aureus*) is a pathogen that has been known to cause severe forms of atopic dermatitis [39,40] through lipoteichoic acid, which is a component of its cell wall and was found to inhibit keratinocyte differentiation and the formation of a healthy barrier by interfering with p63 functions [41].

### 2.2. Effect of Microbes on Cornification and SC Formation

One of the most important components of the skin’s physical barrier is the outermost epidermal layer, the SC, which is the result of specialized keratinocyte differentiation and cornification processes. In this region, dead corneocytes are connected by corneodesmosomes that lie in a lipid-enriched extracellular matrix, which contains ceramides (50%), cholesterol (30–35%), and free fatty acids (10–15%) [13,42]. The corneocytes in the skin are held together by structures called corneodesmosomes. These cells contribute to the skin’s mechanical strength, provide support, and act as a source of hydration for various enzymatic reactions in the SC. The lipid compartment or skin surface lipids (SSL) form a highly ordered three-dimensional structure, called the lipid lamellae, that acts as a permeability, antioxidant, and antimicrobial barrier [35,43,44]. The composition of the SSL changes continuously during keratinocyte differentiation [45], and the ratio of the different components, especially the chain length of the ceramide and the free fatty acid fraction, has important effects on the SC structure and barrier functions [46]. Changes in the compositions can have adverse consequences, such as marked barrier defects secondary to shorter average chain lengths in patients with atopic dermatitis [44,47,48].

The SSLs, in which the corneocytes are embedded, originate from two sources. Epidermal lipids, including ceramides, free fatty acids, and cholesterol, are of keratinocyte origin and are mainly synthesized in the stratum granulosum (SG) layer and released from the lamellar bodies at the SG–SC interface. In parallel with the lipid precursors, the cells secrete hydrolytic enzymes that perform their final processing [49]. Ceramides can be synthesized de novo, but an enzymatic cleavage from sphingomyelin also contributes to a substantial proportion (20–40%) of the total amount. *Staphylococcus epidermidis* (*S. epidermidis*), which is another member of the microbiota, expresses a sphingomyelinase (sph) enzyme that catalyzes the formation of ceramides from lamellar lipids. Together with the keratinocyte-secreted enzymes, sph plays an important role in establishing proper SC composition and function. However, this reaction is also beneficial to the microbe, because the phosphocholine produced is an important nutrient source that can facilitate *S. epidermidis’* colonization of the skin surface [24].

Sebum, another component of the SSL pool, is an amorphous lipid material that is synthesized by sebaceous glands. It contains mostly nonpolar lipids, such as triglycerides, wax esters, and squalene. The density of sebaceous glands is not homogenous, and their uneven distribution contributes to the establishment of dry, moist, and sebum-rich skin areas. This leads to marked differences in the colonizing microbiota and the properties of the cutaneous immune system [4,50,51]. Triglycerides or triacylglycerol lipids are important components of sebum. These molecules are often partially or completely hydrolyzed by host-derived and microbial lipases, resulting in unesterified free fatty acids, mono- and diacylglycerols, and free glycerol. This is a unique feature of the skin of humans compared with many animal species, which were found to have SSLs that contained only the unhydrolyzed mono- and diester waxes [52].

One of the microbes responsible for this enzymatic degradation in human skin is the lipophilic *C. acnes* [25]. The bacterium produces different enzymes, including lipases, proteases, hyaluronidases, acid phosphatases, and sphingomyelinases, and can process different host molecules. Enzymatic processing of triglycerides results in the formation of free fatty acids and glycerol, the latter also being a potential food source for the bacterium [53,54]. Therefore, *C. acnes* actively modifies sebum and, therefore, the SSL composition. In addition, *C. acnes* may directly affect sebaceous gland functions. In hamster sebocytes, the bacterium has been shown to enhance lipogenesis and lipid droplet formation in cell culture supernatants [26]. These effects were partially induced by activation of PPARγ in response to the bacterium itself or its secreted factors [55,56].

These data suggested that cutaneous microbes may regulate SC biogenesis in multiple ways. Resident microbes influence SC formation, enhance keratinocyte differentiation processes, and directly modify the composition of the SSL in their environment through continuous metabolic actions, which appear to be mutually beneficial [22,43].

### 2.3. Cell Junctions Are Also Affected by the Skin Microflora

The SC is a dry, almost impermeable physical structure, but it is not evenly distributed. It is missing from the area of skin appendages (e.g., pilosebaceous units and sweat glands), which serve as molecular transport and contact sites where microbes can easily interact with live skin and immune cells [57]. To maintain the integrity of the body, keratinocytes are tightly packed together in the different skin layers, although their morphological appearance is constantly changing according to their differentiation state. Specialized cell junctions, such as hemidesmosomes, desmosomes, gap junctions, adherent junctions, and tight junctions, are responsible for this tight seal and help in establishing skin barrier functions [13,14,17]. A key step in epidermal differentiation is the formation of the TJ honeycomb, which is a specialized structure with a mesh-like morphology [58]. TJs are particularly important at the sites of different skin appendages, where the other barrier component, the SC, is incomplete [57]. This is the case, for example, in hair follicles, where the SC is present only up to the infundibulum region [59]. In the epidermis, TJ complexes are found only in the second layer of the SG as mature and fully functional sealing structures [60], and these structures are continuous even in the hair follicles, where they are located in the keratinocyte layers bordering the outer environment [61]. TJ pairs form a zipper between adjacent keratinocytes, thereby creating a tight barrier that restricts the free transport of molecules and ions through the paracellular space [62,63]. This regulated movement is referred to as the paracellular transport pathway [64,65].

TJs comprise at least 40 different proteins, the majority of which have transmembrane or adaptor functions [66,67]. The important structural components belong to the protein families claudin (CLDN), occludin, and junctional adhesion molecules, whereas the adaptor molecules, including zonula occludens 1, 2, and 3; cingulin; and afadin link the TJ complex to the cytoskeleton and various signaling cascades to maintain dynamic cell integrity [63,68,69]. The exact composition of TJs can change dynamically in a given tissue, depending on many factors, including age, tissue type, differentiation state, and external and intracellular signals and stimuli [70,71]. In particular, the CLDN content of a cell type is highly regulated spatially and temporally and can change significantly within hours through a process called CLDN switching. Different members of the CLDN family differ in the strength of connection to neighboring cells; therefore, modifying the CLDN content enables TJs to actively adjust their degree of tightness, alter their paracellular permeability properties [64], and adapt to the constantly changing external and internal environment [72]. For this reason, immunostaining of the characteristic TJ proteins has been widely used to gain insight into the integrity of different barriers [73].

Keratinocytes express several different members of the CLDN family, but the most abundant proteins are CLDN1, CLDN4, and CLDN7 [15]. CLDN1 is a key determinant of epidermal barrier function in the human and mouse epidermis, and its levels have been positively correlated with epidermal integrity in rodents [58,74]. CLDN4 establishes a tight linkage between adherent cells, and its decreased expression has been associated with barrier defects. Reduced CLDN1 expression in humans appears to be common in several chronic inflammatory skin diseases, including psoriasis, atopic dermatitis, and rosacea [74,75]. In contrast, CLDN4 often exhibits opposing regulation and is considered a tightening CLDN [72,76], because its upregulation may represent a compensatory mechanism to restore CLDN1-induced loss of barrier functions [10,74,77].

Microbes appear to affect the CLDN content of TJs. In human immortalized keratinocyte cultures and organotypic skin models, *C. acnes* caused changes that were similar to those in CLDN switching and different tissue distribution [28]. Ohnemus and colleagues reported similar findings with *S. epidermidis* [27]. These changes may be secondary to toll-like receptor (TLR) activation, based on findings that in vitro treatment with peptidogly-can enhances TJ barrier functions in cultured keratinocytes and human skin equivalents through activation of TLR2 [78]. This may be an endogenous regulatory mechanism to counteract the barrier-damaging effects of various proinflammatory cytokines that were produced during innate immune and inflammatory activation.

Sebocytes are another cell type in the skin that express the same CLDNs, which possibly function to regulate the permeability barrier but may also be involved in holocrine secretion [79]. Further investigation is required to determine the effect of skin microbes on the expression of these CLDNs and sebaceous gland function.

### 2.4. The TJ and the SC Barriers Are Connected

The two main barrier components, the TJ barrier and the SC barrier, although located in different areas of the skin, are interconnected. This is clearly demonstrated in a small group of patients suffering from NISCH syndrome. Neonatal Sclerosing CHolangitis associated with Ichthyosis (OMIM: 607626) is a novel autosomal recessive disorder in which patients have hyperkeratosis and excessive scaling of the skin [80]. In the early 2000s, its molecular analysis yielded somewhat unexpected results, as a homozygous null mutation of the CLDN1 gene was identified as the causative mutation [81]. These somewhat unexpected findings were explained by the results of recent immunohistochemical studies showing that the SC contains the remnants of TJ structures (e.g., CLDN1 and occludin) that contribute to the cohesion of the horny layer. When TJ barrier defects occur due to reduced expression or a complete absence of CLDN1, compensatory hyperkeratosis and acanthosis attempt to compensate for the loss of TJ barrier functions. Based on this, it can be hypothesized that microbiota-induced changes in CLDN1 expression, e.g., it was described in the case of *C. acnes* and *S. epidermidis* [27,28], may have a complex effect on the permeability barrier, modifying the structure and function of both TJ and SC components [82].

It is interesting to note that the skin of a CLDN1-deficient mouse also shows signs of hyperkeratosis and acanthosis with age, suggesting that similar to humans, abnormal regulation of this protein affects multiple cellular mechanisms in mice [83].

CLD1 has multiple important functions apart from being a central molecule in TJs. It is also important for the regulation of keratinocyte proliferation [84,85], hair growth, and hair cycle progression [61], and is also present in sebocytes [79], which raises the possibility that this molecule is an important regulator of PSU biology. Further investigations should be conducted to unravel its complex function.

## 3. Role of Epidermal Barrier Dysfunction in Acne Pathogenesis

According to the current hypothesis, microbial dysbiosis plays a role in acne pathogenesis, but the involvement of bacterially induced barrier alterations in these processes is currently unclear. *C. acnes* and *S. epidermidis*, which are the most prominent cutaneous microbes, actively regulate the various components of the epidermal barrier. The fact that both, especially *C. acnes*, are implicated in the pathogenesis of acne raises the possibility that bacterium-induced barrier changes may contribute to the development of lesions. One idea was that this cutaneous microbe may have an adjuvant role, although research in the gut strongly suggested that microbial dysbiosis itself led to intestinal barrier defects and chronic immune activation [86,87] Alterations in the interaction between skin cells and the microbiota may be accompanied by marked changes in the chemical barrier caused by the active secretion of various immune effector molecules, including cytokines, chemokines, and antimicrobial peptides. Barrier changes in the PSU wall may lead to increased translocation of irritant and immune-activating molecules to the surrounding tissues, resulting in further inflammation and the formation of papules and pustules (Figure 2).

All the available evidence corroborates the idea that *C. acnes* has a multifaceted effect on keratinocytes. In addition to its role in regulating cell proliferation, differentiation, viability, and innate immune functions, it may also affect cutaneous barrier functions at multiple levels.

## 4. Is It the Quality or Quantity of *C. acnes* That Matters?

In vitro, *C. acnes* has multiple effects on the keratinocyte barrier, and the extent of changes induced may vary depending on the strains and doses used. Low-dose treatments had a barrier-stabilizing effect, whereas a prolonged high-load coculture of certain strains was detrimental to human in vitro cultured keratinocytes [28]. In vivo, the currently favored hypothesis suggests that the bacterial load does not differ significantly between control and acne skin, thereby challenging the notion that the extensive presence of *C. acnes* is solely responsible for acne pathogenesis [88,89]. However, the exact levels of *C. acnes* in the PSU remain uncertain because of limitations in sampling techniques, biofilm formation, and its concentration in deeper regions that are closely attached to the follicle wall and hair shaft [6]. With commonly used sampling techniques, it may not be possible to collect the entire bacterial population from the selected PSU in order to count the number of bacteria present. Thus, researchers should carefully choose the sampling techniques they use to compare the bacterial load in healthy and affected follicles.

Increased sebum secretion creates favorable conditions for this lipophilic bacterium, potentially leading to increased bacterial proliferation during acne lesion development. In addition, anaerobic conditions resulting from plug formation close to the infundibulum may alter the bacterial metabolism and topical distribution, which can lead to microbial dysbiosis and physiological changes in keratinocytes and contribute to follicular hyperkeratinization and comedo formation [90]. Below the forming occlusion, increasingly anaerobic conditions permit further anaerobic bacterial growth and/or metabolism. Moreover, *C. acnes* may produce bacterial molecules and virulence factors, including enzymes, toxins, and short-chain fatty acids (SCFAs), to exacerbate the pathogenic events [91,92].

Among the virulence factors identified are lipases, polyunsaturated fatty acid isomerases, glycosidases, sialidases, and CAMP factors [93]. Their expression under in vivo conditions is not known, nor do we have data on the host factors that modify their expression, and whether and how they contribute to acne pathogenesis. What is known is that dysregulated triglyceride metabolism and immune responses, along with the accumulation of bacterial and host metabolites [91,94], can lead to epidermal barrier changes within the follicle and immune activation in the surrounding tissue. In vitro data seem to corroborate this hypothesis, showing that extensive innate immune and inflammation activation by *C. acnes* leads to decreased claudin-1 expression in cultured keratinocytes. As a result, barrier damage can lead to enhanced paracellular transport through the pore pathway, resulting in the increased passage of large tracer molecules through the keratinocyte monolayer cultures and the epidermis of organotypic skin samples [28].

The exact role of *C. acnes* in vivo in acne pathogenesis remains unclear. The prevalence of specific acne-associated strains, rather than their dose, was previously suggested to be relatively high in affected skin samples. The microbiological and growth characteristics of different strains vary, and some may have a competitive advantage in utilizing nutrients and adapting to anaerobic conditions within the PSU [91]. However, recent evidence suggested that PSUs serve as a bottleneck for *C. acnes* colonization, with a single strain establishing a long-term presence in any given follicle [95]. Although this explains the stability of strain-level community composition in human skin, it raises questions about the precise contribution of acne-associated strains to the disease. Increased susceptibility to inflammation of the follicles that harbor these specific strains remains unclear. Nevertheless, the fact that all currently available effective treatment options have direct or indirect antibacterial or bacteriostatic effects supports the role of a microbial component in the pathogenesis of acne.

## 5. Barrier Damage Appears to Be Common in Various Chronic Inflammatory Diseases

The aforementioned idea of acne pathogenesis shares many features with the model proposed by Cezmi A. Akdis, who suggested common processes in the pathogenesis of several multifactorial diseases of barrier organs (e.g., inflammatory bowel disease, celiac disease, asthma, atopic dermatitis, and chronic rhinosinusitis) [96]. The combined effects of a Westernized lifestyle, microbial dysbiosis, epithelial barrier defects, and inflammation can lead to the development of these conditions [96], including acne [97] (Figure 3). During the pathogenesis of these conditions, a Westernized lifestyle can increase susceptibility to diseases in which bacterial dysbiosis develops due to dysfunctional epithelial barriers. This leads to increased translocation of bacteria or microbial components and results in chronic inflammation in the surrounding sterile tissues. The interplay between these factors that involve barrier alterations and inflammation creates a self-perpetuating vicious cycle and the pathogenesis of chronic inflammatory, atopic, and allergic diseases [96].

Acne may be one of such diseases. Currently, treatment involves a combination of different therapies depending on the disease severity, including topical and systemic molecules with antibiotic and anti-inflammatory effects [98,99]. Importantly, clinicians should bear in mind that *C. acnes* has a complex effect on the skin, including the physical barrier. The use of drugs or treatments that target the epidermal barrier as adjuncts to the currently available mainstream acne drugs would be interesting to investigate in well-designed clinical trials.

## 6. Conclusions

In healthy skin, keratinocytes detect the presence of cutaneous microbes through different pattern-recognition receptor-dependent mechanisms. Among these, the most well-known are the various members of the toll-like receptor (TLR) family, from which TLR2 and 4 are important for the recognition of C. acnes [100,101]. The generated low-grade activation leads to a state of alert in the epidermis and constant training of the resident skin cells, which prepares them for possible pathogenic attacks and helps in maintaining healthy skin functions [102]. Such an effect of the microbiota has been described not only in the skin but also in other barrier organs [103,104]. Through induced signaling events, these microbes promote keratinocyte differentiation [22,37] and the formation and maintenance of a healthy barrier [28]. When pubertal hormonal imbalance alters the conditions in the pilosebaceous units (PSUs), androgen excess leads to increased sebum secretion. This creates a change in the microbiota composition and generates permissive conditions for the lipophilic *C. acnes* [105], resulting in a shift in tissue homeostasis and the formation of bacterial dysbiosis. Increased innate immune and inflammatory activation, together with more extensive keratinocyte activation, characterize this state, to which the increased pathogenic potential of the bacterium may contribute [77,91,106]. Inflammation and certain inflammatory cytokines have negative effects on epithelial barriers [107]. Furthermore, deterioration of the barrier can also cause inflammation [96], and decreased CLDN1 levels can contribute to the hyperproliferation and hyperkeratosis of the infundibular keratinocytes. These events can form a self-perpetuating vicious cycle in the follicle.

The exact mechanism, the sequence of events, and the early steps that set the above processes in motion are not known at this time, but it is believed that hyperkeratosis of keratinocytes in the infundibulum region is a very early step in the development of comedones. We believe that researchers should focus on elucidating barrier changes, especially during the early steps of acne pathogenesis. It would be interesting to systematically investigate the expression levels and changes in barrier molecules and TJ components, especially CLDN-1 levels, to determine if and how they play a role in acne initiation and progression. Accompanying in vitro cell culture experiments may also shed light on whether *C. acnes* plays a role in these early events.

## 7. Final Thoughts

The microbes that live with us were previously thought of as commensal species, adapted to the special conditions provided by our organisms through evolution. However, recent experimental and clinical research has shown that their presence is far from neutral for us. Unraveling the complex interrelationships between our bodies and microbes can contribute to our understanding of the healthy functioning of our bodies and provide a basis for the development of new therapies that are more effective and better tolerated than the current ones.

## Figures and Tables

**Figure 1 ijms-24-15962-f001:**
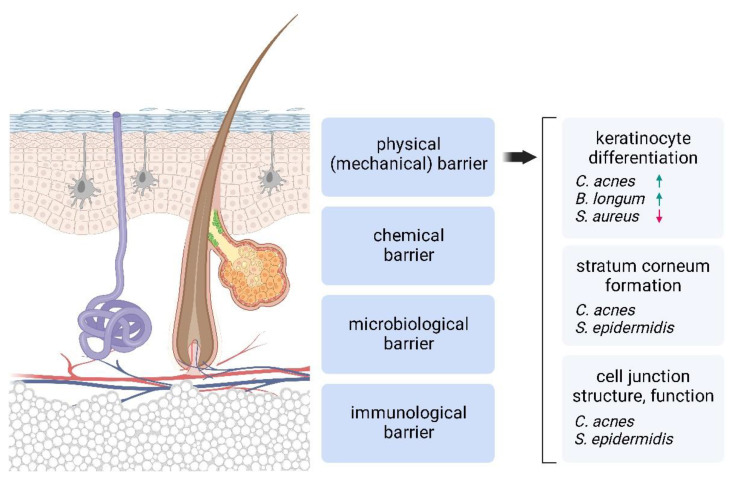
The human skin forms a complex barrier. The skin microbiota affect various properties of this complex interface, even the formation of anatomically intact and functionally mature tissue. The figure shows the bacterial strains with a proven effect on the particular physical barrier component. A green arrow indicates a positive, while a red a negative effect.

**Figure 2 ijms-24-15962-f002:**
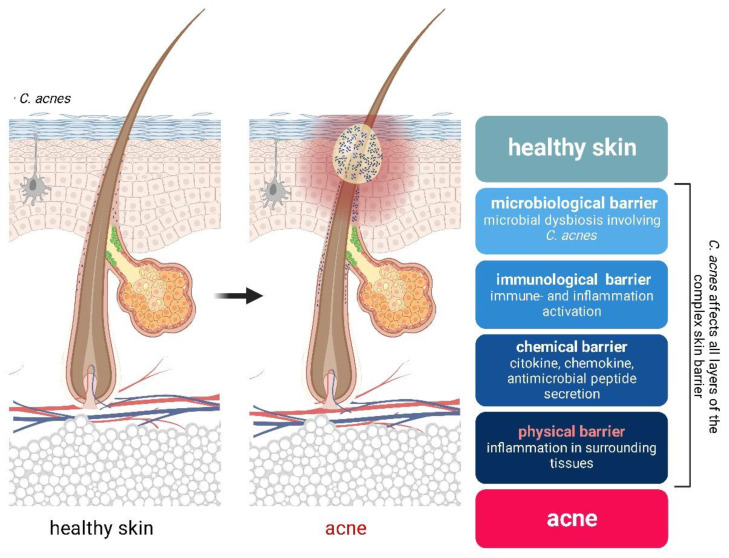
In healthy skin, the skin cells and microbiota, especially the bacterium *C. acnes*, are in equilibrium. During puberty, microbial dysbiosis occurs, causing significant changes at all levels of the complex skin barrier, all of which play a major role in the pathogenesis of acne.

**Figure 3 ijms-24-15962-f003:**
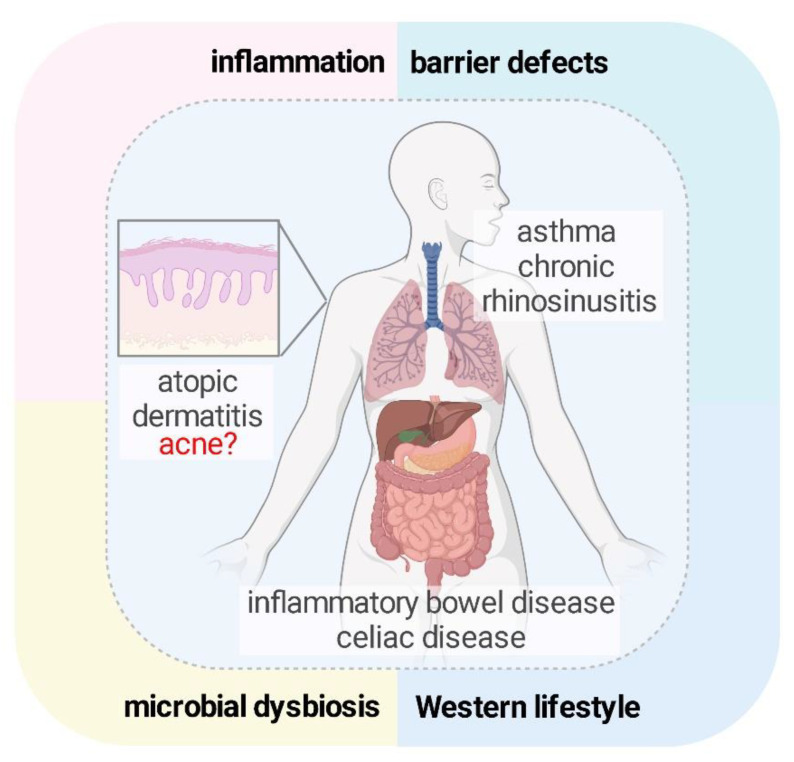
The combined effect of many factors, such as barrier defects, is common in the pathogenesis of common chronic diseases of barrier organs, including acne.

**Table 1 ijms-24-15962-t001:** Different members of the skin microbiota have a complex effect on various components of the mechanical barrier.

Barrier Components/Processes Involved in Physical Barrier Formation	Possible Effect of the Microbiota
keratinocyte differentiation processes	role in the development of mature skin tissues -aiding the early, neonatal adaptation processes [20] -regulation of keratinocyte differentiation processes by affecting the gene expression of EDC genes [21,22,23]
SC formation	modification of SSL composition [24] modification of sebum composition [25,26]
cell junctions	modification of TJ structure by affecting the expression of its components [27,28]

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
