# Peer review of "Are the Cutaneous Microbiota a Guardian of the Skin’s Physical Barrier? The Intricate Relationship between Skin Microbes and Barrier Integrity"

_ijms, 2023, doi:10.3390/ijms242115962_

Round 1
Reviewer 1 Report
Comments and Suggestions for Authors
Please see this reviewer's comments in the attached document.

One consideration is to avoid using "our" in discussions of the skin and it's function, e.g., line 28.
Author Response
Answers to Reviewer 1.
First of all, we would like to thank the Reviewer for his/her thorough assessment of our paper. We believe that the revised manuscript based on his comments has been greatly improved compared to the original version.
All introduced changes and revised parts in the new version of the manuscript are marked in yellow.
Please find below our point-by-point responses to the questions and comments raised.
- The manuscript structure discusses many aspects of the skin barrier properties and function, as well as the “mechanical” barrier. The formation and structure, including appendages, are covered and convey the complexity. However, the aspects of microbes, microbiome and Cutibacterium acnes are interspersed throughout the numbered sections making creating awkward transitions and lacking a logical flow of information. Consider revising to first discuss the structure and function of the skin and appendages, with the overlay of the developmental timeline, and then describing how the microbiota and skin barrier influence each other. Such a clarification will help the readers understand the interaction.
Our main aim is not to describe the structure, function and development of the skin, as there are many excellent summaries and textbooks on this subject. In this manuscript, we rather focus on the role of the microbiota, and in particular Cutibacterium acnes (C. acnes), on some of the known components of the mechanical barrier. Therefore, each chapter contains a brief summary of the properties of the barrier component, supplemented by the effect of the microbes on these. For this reason, we feel it is useful to retain the original structure and rationale.
- In total, the paper appears to be about microbes in general rather than Cutibacterium acnes specifically. Consider changing the title to match the paper contents.
We agree with the reviewer that in some chapters the known effects of other microbes besides C. acnes are analysed, and have therefore changed the title of the manuscript to The skin microbiota has a complex effect on the mechanical skin barrier properties.
- When review papers are cited for specific statements, check the original sources within them to ensure that the information is correctly interpreted and stated.
We are trying to make sure that the statements that are quoted are correctly interpreted by us and are included in the text.
Specific Comments
- Line 28 – expand to include protection from what, regulation or what and sensation of what.
The sentence has been changed. Please refer to the revised version of the manuscript.
- Line 30 suggests that “mechanical” barrier is interchangeable with “physical” barrier. Please clarify. Consider also that the phrase “mechanical barrier” is often taken to mean biomechanical in the sense of elastic properties.
In the original manuscript, the two terms are used in a similar sense. The reason is that these terms are also found in the literature to describe the property of the skin that is physically able to prevent the passage of various substances and particles through it. The term mechanical barrier is mentioned here in a similar sense (Eyerich S, Eyerich K, Traidl-Hoffmann C, Biedermann T. Cutaneous Barriers and Skin Immunity: Differentiating A Connected Network. Trends Immunol. 2018;39(4):315-327. doi:10.1016/J.IT.2018.02.004). To make it less confusing, we use now physical barrier throughout the manuscript.
- Lines 31-33 are confusing. What is the intended point? Perhaps “well-being” is too general to be properly interpreted.
To be more specific, we have rephrased the highlighted sentence, which now reads: The interconnectedness of these barriers and the importance of a tightly regulated, balanced interface between us and the environment is critical to maintaining homeostatic conditions in our bodies.
- Lines 49-51 imply that the impact of microbes on the physical barrier are limited. There are a number of studies that describe the interaction between microbes and the barrier. It may not be possible to separate the physical barrier of the skin from the immunological and chemical barriers as the physical structure of corneocytes embedded in a lipid matrix serve an 2 immunological function. Consider not limiting the review to the mechanical barrier as defined in the manuscript.
We agree that it is rather difficult, in some cases, to separate the physical, chemical, immunological, and microbiological barrier functions of the skin, as these are highly interconnected. Our article aims to focus on the effect of the microbiota on the physical barrier, the development and the anatomical structure of the known barrier components, the stratum corneum and the tigtht junction belt. We chose this topic because there are many excellent reviews and other articles e.g. on how microbes affect the immunological and chemical barrier, but the physical barrier is somehow overlooked. We believe that this topic is interesting in its own right and deserves more attention than is currently being paid to it.
- Lines 59 – 61 refer to dead corneocytes and the process of dying. The keratinocytes transition when the lose their ability to differentiate. Consider checking the cited references and refining this statement.
This section has been revised. Please see the corresponding text.
- The paragraph with lines 62-69 discusses the viable epidermis. Is the point of this paragraph to explain that the mechanical barrier is derived from living layers?
In part, the answer to the above question is yes. However, we would like to emphasize that proper regulation of keratinocyte differentiation events is critical for the formation of a structurally and functionally mature epidermal barrier. To further emphasize this, we revised the corresponding text.
- The statement in lines 74-76 beginning with “full-term infants already have….” is misleading because the full-term newborn stratum corneum has a low TEWL and serves to protect the infant from water loss. The authors are correct that the skin barrier continues to mature.
Thank you for this remark, the corresponding text has been modified accordingly.
- Are there citations for lines 77-80? The statement about maintaining functionally mature tissues later in life appears to be speculative.
This idea is speculative, we have not yet found a reference in the literature. We believe that we have indicated this by the wording of the sentence in question. In addition, we also slightly changes it to emphasize the hypothetical nature of this idea: The timing of these events raises the possibility that, in addition to assimilation to a new environment, active contact with microbes may support early neonatal development and the adaptation processes of the skin and may also help maintaining anatomically and functionally mature tissues later in life.
- In line 82, specify the type of cutaneous microbiota. Should this be commensal?
Even though the term commensal is still generally used in the literature, because of the proven two-way interaction that exist between the skin and its colonized microbes, we try to refrain from the use of it. In this part, we have referred to the resident cutaneous microbiota. This term is now included in the sentence in question.
- Line 86-87 is misleading because the basket weave appearance is sometimes an artifact of the histological processing method. The stratum corneum is about 16 cell layers thick with a lipid bilayer structure between the corneocytes. The cells are held together by desmosomes to maintain a strong structure. The SC can be separated from the epidermis and is an intact layer.
Thank you for the remark regarding the SC structure. We have deleted the part referring to the basket weave appearance.
- The statement in line 89 implies that the SC structure is evidence that microbes impact it. Check the reference as the statement is misleading.
We have changed the term "clearly demonstrates" to "suggests" as we believe that the available animal data are indicative of the effects of cutaneous microbes on SC structure and function.
- Clarify the information in lines 90-97. Were there two groups of mice in the work cited? How would the information in germ-free condition and involving mice be relevant to human skin? Is there corresponding evidence in humans? Please clarify the point and how it might apply to human skin and human skin disease.
- For the information in lines 98-107, the cited studies were done on keratinocytes. Consequently, the statement in lines 105-107 should be revised as it is over-reaching and speculative as written.
We have changed the phrasing of the sentence as follows: These data suggest that members of the cutaneous microbiota may have some effect on the establishment and maintenance of a mature epidermis and the development of healthy cutaneous barrier functions.
- Another comment on the paper’s organization is for section 2.2 (beginning in line 116). The first paragraph discusses structure with the role of microbes starting in line 137. Rather than going back and forth between structure and “microbes”, consider reorganizing the paper to begin with structure to set the basis for discussions about microbiomes.
Thank you for your comment. We agree that we do not aim to follow the classic textbook style of organisation in our article. However, we believe that this perhaps unconventional style better serves our purpose. Therefore, in each section we briefly introduce a particular barrier component and in the second part we discuss how microbes can affect this particular feature.
- In line 170, please clarify what is meant by “.it is not evenly distributed….” Also state why “interruption” with appendages is important to the point being made.
I think this is explained in the following sentence, which says that at the sites of various skin appendages (pilocebaceous units, sweat glands) this structure is missing or not continuous. We wanted to emphasize the importance of the other components of the barrier, the cellular junctions that are present at the sites of these structures.
- In lines 228-230, consider revising the statement regarding the belief that TJ barriers and SC are independent. That is, provide a reference for the statement that they are independent as others would likely argue that they are interdependent.
We have revised this section, which now reads: The two main barrier components, the TJ barrier and the SC, although located in different areas of the skin, are interconnected. This is clearly demonstrated in a small group of patients suffering from NISCH syndrome. Neonatal Sclerosing CHolangitis associated with Ichthyosis (OMIM: 607626) is a novel autosomal recessive disorder in which patients have hyperkeratosis and excessive scaling of the skin [79].
- For the statements starting in line 252, please include citations or elaborate to support the statement further.
Thank you for this remark. We have included some additional references on the complex role of CLDN1 apart from the regulation of TJ functions.
In the same section, beginning in line 261, the authors state that healthy skin keratinocytes might detect microbes, etc. Perhaps this section should be set apart as a conclusion from the review so as to clarify what is suggested by the current data along with what needs to be done in the future. It would be very helpful if the authors would clearly state future directions so readers can see what is “possible” (perhaps speculation) and what needs to be done to confirm.
We have reorganized some parts of the manuscript as suggested by the reviewer (see the corresponding text in Section 6) and included some thoughts on how we think further studies can improve our understanding of the effect of C. acnes on the physical barrier.
Reviewer 2 Report
Comments and Suggestions for Authors
The manuscript entitled “CUTIBACTERIUM ACNES HAS A COMPLEX EFFECT ON THE MECHANICAL SKIN BARRIER PROPERTIES“ could be a valuable review which presents this topic, but it is necessary to improve (expand) its content.
- The title (and the article) is based on data on Cutibacterium acnes, but the manuscript involves a little data on this microorganism. Also, the title is too general.
-Figure 1 is too general, there are not specific names of microorganisms involves in this review.
-Sentence: “Several types of specialized cellular constructs or intercellular junctions (desmosomes, hemidesmosomes, gap, adherent and tight junctions) play important roles in performing these tasks [ 13–17]”- Which tasks?
-Sentence: “Many of the loci that have been identified belong to the epidermal differentiation complex, which functions in barrier formation [ 28] and skin and epidermis development [ 27]” -Loci for genes?
-I suggest to add one table or a visual presentation of key data mentioned in the manuscript.
-In the sentence, the reference is missing “The aforementioned idea of acne pathogenesis shares many features with the model proposed by Cezmi A. Akdis, who suggested common processes in the pathogenesis of several multifactorial diseases of barrier organs (e.g., inflammatory bowel disease, celiac disease, asthma, atopic dermatitis, and chronic rhinosinusitis)“.
-The roles of microorganisms in mechanical barrier (sections 2.1. - 2.4. ) could be presented in one table, with their key features
- Section 2.4. The association of this section with Cutibacterium acnes should be mentioned
-Some sentences could be more clearly presented:
*Based on the available evidence, we thought that in healthy skin, keratinocytes detect the presence of cutaneous microbes through different pattern recognition receptor-dependent mechanisms.“
*Furthermore, deterioration of the barrier can also cause inflammation [ 92], and decreased CLDN1 levels can contribute to the hyperproliferation and hyperkeratosis of the infundibular keratinocytes.“
-There are some typos in the sentences: „The exact mechanism, the sequence of events, and the early steps that set the above processes in motion are n not known at this time, but it is believed that hyperkeratosis of keratinocytes in the infundibulum region is a very early step in the development of comedones. „
It would be interesting to investigate whether barrier damage, specifically the reduction of CLDN-1 levels, plays a role in acne initiation, and whether or not C. acnes plays a role in these early events?
- Please improve the style, when possible.
-When talk about skin barrier, it is needed to mention contact dermatitis
-What is PSU? There is no full name.
- I suggest to mention resident bacteria and transient bacteria
- Also, some ADDITIONAL DATA could be mentioned: For example, some useful data from the two articles:
-the article by Mayslich, C., Grange, P. A., & Dupin, N. (2021). CUTIBACTERIUM ACNES AS AN OPPORTUNISTIC PATHOGEN: AN UPDATE OF ITS VIRULENCE-ASSOCIATED FACTORS. Microorganisms, 9(2), 303. https://doi.org/10.3390/microorganisms9020303 ):The genus Cutibacterium acnes (C. acnes, formerly known as Propionibacterium acnes or P. acnes, see below) is a commensal lipophilic Gram-positive bacterium. C. acnes is described as diphtheroid or coryneform because it is rod-shaped and slightly curved with a width of 0.4 to 0.7 µm and length of 3 to 5 µm. Anaerobic bacteria are characterized by their inability to grow on solid media in the presence of atmospheric oxygen. However, C. acnes is considered an aerotolerant anaerobe because it possesses enzymatic systems able to detoxify oxygen, allowing it to be sustained on the surface of the skin [1].Following its isolation [2], C. acnes was first included in the genus Bacillus as Bacillus acnes, and then in the genus Corynebacterium as Corynebacterium acnes or “anaerobic corynebacteria” because of its morphology. Based on its ability to produce propionic acid via its anaerobic catabolism, it was then assigned to the genus Propionibacterium, subsequently renamed Cutibacterium. Genus Cutibacterium belongs to a branch of Actinobacter and can be split into two groups, one containing the so-called “classic or dairy” species, bringing together saprophytic species isolated from non-human-pathogenic dairy products, and the other containing commensal “skin” species, most found on the surface of human skin. Classic species, such as Propionibacterium freundenreichii, which is essential for the ripening of Swiss cheeses, or Propionibacterium acidipropionici, known for its beneficial effects in the bovine rumen, have been studied in considerable detail due to their importance to the agri-food industry. By contrast, the pathophysiology of cutaneous species is less well understood.
- the article by Kumar B, 2016 (NEW INSIGHTS INTO ACNE PATHOGENESIS: EXPLORING THE ROLE OF ACNE-ASSOCIATED MICROBIAL POPULATIONS): The pathogenic life cycle of bacteria is mediated by virulence genes encoding virulence factors within their pathogenic islands. The virulence genes, unlike house-keeping genes, are characterized by the production of toxins, adhesins, invasions, or other types of factors, present preferably in the pathogenic microorganisms.9, 10 These products are directly involved in the pathological damage to the host by promoting interaction between the host and organism and also by damaging and degrading the host tissues. For instance, camp5, gehA, tly, sialidases, neuraminidases, and endoglycoceramidases are some of the virulence factors of Propionibacterium acnes which causes acne vulgaris.11 Lipases, fatty acid modifying enzyme, polysaccharide intercellular adhesion (PIA), and poly-glutamic acid are the virulence factors in Staphylococcus epidermidis.12, 13 Adhesins, fibronectin binding protein (FnBp)-A, FnBP-B, proteases, lipases, and hyaluronidases are the virulence factors in Staphylococcus aureus.14, 15 Thus, each pathogen follows its own pathogenic strategy, with a diverse and unique set of genes/factors operating in a concerted manner to cause disease in the host.
Author Response
Answers to reviewer 2.
To start, we'd like to express our gratitude to the Reviewer for their comprehensive evaluation of our paper. We are confident that the revised manuscript, which incorporates the Reviewer's feedback, represents a substantial enhancement over the initial version.
All modifications and updated sections in the new manuscript are highlighted in yellow. You will also find our detailed responses to the questions and comments raised below.
The manuscript entitled “CUTIBACTERIUM ACNES HAS A COMPLEX EFFECT ON THE MECHANICAL SKIN BARRIER PROPERTIES“ could be a valuable review which presents this topic, but it is necessary to improve (expand) its content.
- The title (and the article) is based on data on Cutibacterium acnes, but the manuscript involves a little data on this microorganism. Also, the title is too general.
We have revised the title and changed to the following: Is the cutaneous microbiota a guardian of the skin's physical barrier? The intricate relationship between skin microbes and barrier integrity
- Figure 1 is too general, there are not specific names of microorganisms involves in this review.
We have modified Figure 1, accordingly.
- Sentence: “Several types of specialized cellular constructs or intercellular junctions (desmosomes, hemidesmosomes, gap, adherent and tight junctions) play important roles in performing these tasks [ 13–17]”- Which tasks?
In this sentence we referred to the close connection, described in the previous sentence. To clarify this, we have changed the part as follows: “Several types of specialized cellular constructs or intercellular junctions (desmosomes, hemidesmosomes, gap, adherent and tight junctions - TJs) play important roles in provid-ing such close connection [13–17].”
- Sentence: “Many of the loci that have been identified belong to the epidermal differentiation complex, which functions in barrier formation [ 28] and skin and epidermis development [ 27]” -Loci for genes?
Thank you for pointing out this mistake. We also further revised the sentence. Correctly: “Many of the genes that have been identified belong to the epidermal differentiation complex (EDC, 1q21), including filaggrin, loricrin, involucrin, small prolin-rich porteins, S100 proteins, which functions in barrier formation [ 28] and skin and epidermis development [ 27]”
- I suggest to add one table or a visual presentation of key data mentioned in the manuscript.
We have prepared one figure (Figure 3) and modified Figure 1 according to the reviewer's suggestions. We believe that together the modifications provide a visual representation of all the key elements of our manuscript.
- In the sentence, the reference is missing “The aforementioned idea of acne pathogenesis shares many features with the model proposed by Cezmi A. Akdis, who suggested common processes in the pathogenesis of several multifactorial diseases of barrier organs (e.g., inflammatory bowel disease, celiac disease, asthma, atopic dermatitis, and chronic rhinosinusitis)“.
The appropriate reference has been placed here.
- The roles of microorganisms in mechanical barrier (sections 2.1. - 2.4. ) could be presented in one table, with their key features
We have included a table (Table 1) with the corresponding data, as suggested.
- Section 2.4. The association of this section with Cutibacterium acnesshould be mentioned
We have revised the suggested sentence, which now reads: “Based on this, it can be hypothesized that microbiota-induced changes in CLDN1 expression, as in the case of C. acnes and S. epidermidis (Ohnemus; 2008; Bolla, 2020), may have a complex effect on the permeability barrier, modifying the structure and function of both TJ and SC components [81].”
- Some sentences could be more clearly presented:
*Based on the available evidence, we thought that in healthy skin, keratinocytes detect the presence of cutaneous microbes through different pattern recognition receptor-dependent mechanisms.“
This section has been modified as follows: In healthy skin, keratinocytes detect the presence of cutaneous microbes through different pattern recognition receptor-dependent mechanisms. Among these the most well-known are the various members of the Toll-like receptor (TLR) family, from which TLR2 and 4 are important for the recognition of C. acnes (Pivarcsi, 2003; Nagy; 2005)
*Furthermore, deterioration of the barrier can also cause inflammation [ 92], and decreased CLDN1 levels can contribute to the hyperproliferation and hyperkeratosis of the infundibular keratinocytes.“
We have revised the marked sentence, which now reads: “Furthermore, deterioration of the barrier can also cause inflammation [ 92] and in vitro, reduced CLDN1 levels can contribute to hyperproliferation (De Benedetto, 2011). Increased proliferation and hyperkeratosis of infundibular keratinocytes are characteristic features of acne pathogenesis and are among the earliest detectable changes.”
- There are some typos in the sentences: „The exact mechanism, the sequence of events, and the early steps that set the above processes in motion are n not known at this time, but it is believed that hyperkeratosis of keratinocytes in the infundibulum region is a very early step in the development of comedones. „
It would be interesting to investigate whether barrier damage, specifically the reduction of CLDN-1 levels, plays a role in acne initiation, and whether or not C. acnes plays a role in these early events?
Thank you for pointing out these mistakes.
- Please improve the style, when possible.
-When talk about skin barrier, it is needed to mention contact dermatitis
We have added contact dermatitis to the list of diseases in which defects in the skin barrier play a pathogenic role.
-What is PSU? There is no full name.
Thank you for pointing this out, the full name was included when the term first appeared (page 2. line 48).
- I suggest to mention resident bacteria and transient bacteria
This was included in the manuscript (page 2. lines 46-48).
-Also, some ADDITIONAL DATA could be mentioned: For example, some useful data from the two articles:
-the article by Mayslich, C., Grange, P. A., & Dupin, N. (2021). CUTIBACTERIUM ACNES AS AN OPPORTUNISTIC PATHOGEN: AN UPDATE OF ITS VIRULENCE-ASSOCIATED FACTORS. Microorganisms, 9(2), 303. https://doi.org/10.3390/microorganisms9020303 ):
We have included a brief description of the C. acnes virulence factors on page 9. lines 332-336, as suggested.
Round 2
Reviewer 1 Report
Comments and Suggestions for Authors
The authors have addressed all of this reviewer's comments. The revised manuscript is substantially improved. The addition of the new Figure 2 is very helpful to readers. I have pnly one comment and it is minor In Table 1, consider putting a line between each of the major barrier components sections so the reader knows which "possible effect" goes with each component.
Author Response
Once again, we would like to thank the Reviewers for his/her suggestions and criticisms, which have greatly contributed to improving the quality of the manuscript.
The changes requested in Table 1 have been made.
We hope that you find the manuscript in its current form to be of an appropriate standard for publication in IJMS.
Reviewer 2 Report
Comments and Suggestions for Authors
The authors changed the text according the recommendations. So, the manuscript has improved.
Author Response
We want to express our gratitude to the Reviewer for his/her valuable suggestions and constructive criticisms.